# Saliva and Saliva Extracellular Vesicles for Biomarker Candidate Identification—Assay Development and Pilot Study in Amyotrophic Lateral Sclerosis

**DOI:** 10.3390/ijms24065237

**Published:** 2023-03-09

**Authors:** Sebastian Sjoqvist, Kentaro Otake

**Affiliations:** Neuroscience Drug Discovery Unit, Takeda Pharmaceutical Company Limited, 26-1, Muraoka-Higashi 2-chome, Fujisawa 251-8555, Japan

**Keywords:** saliva, extracellular vesicles, biomarkers, amyotrophic lateral sclerosis

## Abstract

Saliva is gaining increasing attention as a source of biomarkers due to non-invasive and undemanding collection access. Extracellular vesicles (EVs) are nano-sized, cell-released particles that contain molecular information about their parent cells. In this study, we developed methods for saliva biomarker candidate identification using EV-isolation and proteomic evaluation. We used pooled saliva samples for assay development. EVs were isolated using membrane affinity-based methods followed by their characterization using nanoparticle tracking analysis and transmission electron microscopy. Subsequently, both saliva and saliva-EVs were successfully analyzed using proximity extension assay and label-free quantitative proteomics. Saliva-EVs had a higher purity than plasma-EVs, based on the expression of EV-proteins and albumin. The developed methods could be used for the analysis of individual saliva samples from amyotrophic lateral sclerosis (ALS) patients and controls (*n* = 10 each). The starting volume ranged from 2.1 to 4.9 mL and the amount of total isolated EV-proteins ranged from 5.1 to 42.6 µg. Although no proteins were significantly differentially expressed between the two groups, there was a trend for a downregulation of ZNF428 in ALS-saliva-EVs and an upregulation of IGLL1 in ALS saliva. In conclusion, we have developed a robust workflow for saliva and saliva-EV analysis and demonstrated its technical feasibility for biomarker discovery.

## 1. Introduction

Saliva is gaining increasing attention as a source for biomarker research. The non-invasive and patient friendly collection are two major benefits. The collection is simple and does not require specialized training. Additionally, in contrast to blood, saliva does not coagulate and requires less post-collection manipulation [1]. A good demonstration of the usefulness of saliva is testing for coronavirus disease 2019 (COVID-19). Extensive testing was recognized early as an important cornerstone of the public health response to the pandemic. Nasopharyngeal sampling was initially the most commonly used COVID-19 test, but these require trained personnel and patient discomfort made the sampling difficult in certain patient groups (for example, children). However, more recently, saliva tests have emerged. The sample collection is more convenient, with less discomfort and without the need of trained staff. A recent systematic review and meta-analysis revealed that there was no difference in sensitivity or specificity between saliva based and nasopharyngeal swab nucleic acid amplification testing for COVID-19 [2].

Saliva has also been investigated as a source of biomarkers for oral cancer [3], periodontitis [4], epilepsy [5] and Alzheimer’s disease [6]. Non-invasive biofluid collection could also increase the frequency of sampling, and patients could be likely to agree to provide samples daily or even several times per day, if the collection procedure is simple enough. This could enable invaluable information gathering regarding the kinetics of biomarkers. Further, the increased number of data points could be important for artificial intelligence approaches, where data availability is a common hurdle for success [7]. Additionally, large-population screenings could be greatly facilitated if sampling could be done by everyone.

However, saliva has disadvantages because the samples can be contaminated by viruses, bacteria and food debris [8]. Such potential contamination is important to take into consideration when developing saliva-based molecular biomarkers. Examples of measures that can be taken are defined mouth washing protocols, restrictions of diet prior to sample collection or making sure that the biomarker candidate is specific to humans. Another drawback is that the protein concentration of the target biomarker might be considerably lower compared to plasma concentrations, which might make it technically difficult to quantify accurately and robustly [1]. Additionally, collecting saliva from research animals such as rodents is possible but more technically difficult than blood, and might represent a challenge from a translational medicine perspective [9].

Extracellular vesicles (EVs) are particles sized 50–10,000 nm that are released from all cells [10]. They play essential roles in cellular communications and can transfer bioactive molecules such as growth factors between cells [11]. EVs have been isolated from most bodily fluids, including urine [12], plasma [13], cerebrospinal fluid (CSF) [14], semen [15], saliva [16,17,18], etc. The exact composition and origin of saliva-EVs remain unknown, but salivary glands, oral mucosal cells (epithelial cells, fibroblasts, etc.) and oral flora likely contribute [19]. EVs contain a fingerprint of the parent cells and are therefore being extensively investigated as “liquid biopsies”. Gai et al. demonstrated that saliva-EVs contain miRNAs that could potentially serve as biomarkers for oral squamous cell carcinoma [17]. Similarly, Li et al. reported a multicenter study evaluating a biomarker signature based on small RNAs in saliva exosomes to diagnose esophageal carcinoma, with a sensitivity of 90.5% and a specificity of 94.2% [20]. Being able to gather molecular information of organs that are difficult to biopsy (e.g., the brain) in a non-invasive manner could have an incredible impact for understanding underlying pathogenesis, early detection of disease, drug effects, etc.

In this study, we aimed to develop methodologies for analyzing saliva and saliva EVs using proteomic techniques. We developed a workflow using pooled, large volume, human saliva samples and proceeded with a pilot, technical feasibility study using low volume individual samples from amyotrophic lateral sclerosis (ALS) patients and controls. Figure 1 shows the study overview.

## 2. Results

### 2.1. Assay Development Using Pooled Saliva

Pooled, commercially available saliva from healthy donors was used for assay development. The saliva was cleared using differential centrifugation (300× *g* and 3000× *g*) and EVs were isolated using exoEasy, followed by a buffer-exchange step using 100 kDa ultrafiltration. Three independent isolations were performed. The total protein concentration in saliva was 486 ± 10.4 µg/mL and the total amount of protein in EVs isolated from one mL of saliva was 1.6 ± 0.9 µg, representing a reduction of approximately 99.5% (Figure 2a). We measured albumin as an indicator of non-EV contaminants, as recommended by the International Society for Extracellular Vesicles 2018 guidelines [21]. The albumin concentration in saliva was 16.3 ± 2.1 µg/mL, while EVs isolated from one mL of saliva contained 26 ± 8.7 ng/mL, representing a reduction of 99.8% (Figure 2b). Transmission electron microscopy revealed spherical shaped vesicles (Figure 2c,d). We used nanoparticle tracking analysis to measure the size and concentration of the EVs. The EV-isolates showed a rather heterogenous particle population with several peaks (Figure 2e). On average, 6 ± 1 × 10^8^ particles could be isolated from one mL of saliva and the mean particle size was 309 ± 41 nm (Figure 2f).

Next, we evaluated the feasibility of two technically different proteomic approaches, proximity extension assay (PEA) and label-free quantitative proteomics. We analyzed EVs and saliva by PEA using the panel “Olink Target 96 Cardiovascular III(v.6113)”. This panel was chosen because some analytes have been reported to be related to ALS (e.g., YKL-40 [22], CHIT1 [23], MCP-1 [24]) and we have previously used this panel for CSF-based biomarker discovery for ALS [25]. For neat (non-diluted) EVs, 85–88 proteins could be detected, while for neat saliva 86–88 proteins could be detected. The number of detected proteins decreased with higher dilutions, confirming the sensitivity (Figure 3a). A multi scatter plot indicates a high intra-group correlation for both EVs and saliva (Pearson correlation 0.999) and a moderate correlation between the two groups (Pearson correlation 0.845) (Figure 3b). To visualize differentially expressed proteins between the two groups, a volcano plot was produced. Fifty-four proteins were upregulated in saliva, while fifteen were upregulated in saliva-EVs and twenty-three had a similar expression between the two groups (Figure 3c).

Saliva-EVs were compared to plasma-EVs using label-free quantitative proteomic analysis. Three independent isolations for each sample type were used. In this analysis, we did not include whole saliva (in contrast to PEA above) since it is difficult to compare samples with too different composition. Among the detected proteins, two proteins that are considered EV-contaminants according to International Society for Extracellular Vesicles (ISEV) 2018 guidelines [21], APOA1 and albumin, were identified. While there was no difference in APOA1 between the groups, albumin was greatly reduced in saliva-EVs compared to plasma-EVs (Figure 4a). We could also detect several EV-markers belonging to “transmembrane proteins” and “cytosolic proteins recovered in EVs”, as recommended in the ISEV 2018 guidelines [21]. Multiple markers had a significant increase in saliva-EVs compared to plasma-EVs: CD9, CD55, CD63, CD81, RHOA, ANXA1 and HSPA8.The remaining protein showed a similar expression between the groups (Figure 4b,c). A multi scatter plot demonstrated that the intra-group correlation was higher for saliva-EVs compared to plasma-EVs (Pearson correlation 0.856 and 0.809, respectively) and that the correlation between the two groups was low (Pearson correlation 0.436) (Figure 4d). Unsupervised analyses could successfully divide the samples into each corresponding group, as demonstrated by hierarchical clustering and principal component analysis (PCA) plots, indicating clear inter-group differences (Figure 4e,f). A volcano plot was used to demonstrate differently expressed proteins. There were 295 proteins upregulated in saliva-EVs, 326 upregulated in plasma-EVs and 848 without any significant difference between the groups. We used the “ExoCarta” database [26] (accessed on 13 February 2023) to compare identified proteins to the top 100 “Exosome proteins” and found that 66 were detected in our samples while 34 were not (Appendix A), which suggests a successful enrichment of EVs.

Finally, the input material based on micrograms of total protein required was evaluated. In 2 µg of total protein, we could detect 1537 unique proteins and only five more were detected when analyzing 20 µg of input material. However, when the input amount was reduced to 0.5 µg, the number of detected proteins was reduced to 1456 (Figure 4h). Therefore, 2 µg of total protein was determined to be the minimum required amount of saliva-EV protein for future measurements of individual samples from patients or controls.

### 2.2. Feasibility Study Using ALS and Control Samples

Saliva samples from ALS patients and matched controls were used for a feasibility study. The donor details and clinical characteristics are presented in Table 1.

Saliva from control (4.6 ± 0.5 mL) and ALS (2.9 ± 0.4 mL) participants was used for the study. Four control samples and two ALS samples contained macroscopically visible contamination (gel, brown particles, or fibers) (Figure 5a). Approximately 30% of the starting volume was lost during the clearing step, and the volume of samples with visible contamination lost more than others (Figure 5b). There was a large variability in the protein concentration for both groups: control samples had 892 ± 514 and ALS had 1341 ± 847 µg/mL. There was no statistically significant difference between the groups in terms of protein concentration and they were in a similar range as the pooled samples used for assay development (Figure 5c). Similarly, there was no difference in the amount of EV-protein that could be isolated from one mL of saliva (yield); 7.4 ± 2.9 and 7.8 ± 4.7 µg for control and ALS-samples, respectively (Figure 5d). On average, the isolated EV-proteins represented 1 ± 0.5 and 0.7 ± 0.4% of the total protein in saliva for control and ALS-samples, respectively (Figure 5e), again without difference between the groups and in similar range as the pooled samples. The total amount of proteins that could be isolated from each sample ranged from 42.6 µg to 5.1 µg (Figure 5f). Based on the minimum value, 4 µg of each sample was selected to be used for label-free quantitative proteomics.

There was no clear distinction between the two groups of saliva-EVs based on PCA, and there was no indication that contamination affected the analysis (Figure 6a). There were no significantly differentially expressed proteins, but a trend for a decrease in ZNF428 in ALS saliva-EVs was observed (Figure 6b,c). Similarly, for the saliva samples, there was no clear clustering in the PCA plot (Figure 6d) and no differentially expressed proteins, but a trend for an increase in IGLL1 in ALS patients was found (Figure 6e,f). Peptide information regarding ZNF428 and IGLL1 is included in the Appendix A.

## 3. Discussion

A workflow for the analysis of saliva and saliva-derived EVs was developed in this study, and its technical feasibility confirmed using individual samples from ALS patients and controls. ISEV’s “Minimal information for studies of extracellular vesicles”-guidelines for the characterization of the EVs were followed. This includes the analysis of particle size and concentration, visualization, and the detection of “positive” and “negative/contaminating” EV-proteins. Recently, the number of commercially available kits for extracellular vesicles has increased drastically, each with advantages and disadvantages. In this study, we decided to use exoEasy based on our previous experience with this kit, and the fact that it is less labor-intense and requires fewer manual steps (which introduces variability) than other kits that we have tried. In a recent study, exoEasy was systemically compared to other kits and it was concluded that the kit was well suited for plasma samples, with a high number of EV-proteins and few non-EV proteins [27].

Albumin was used as a proxy for contaminating proteins and a decrease of over 99.8% compared to cleared saliva was noted, which is similar to previous results from CSF-EV-isolation [14]. A buffer-exchange step after the EV-isolation was used in order to replace exoEasy’s proprietary “Buffer XE” with PBS. Based on our previous experience, “Buffer XE” might interfere with antibody binding and CSF-EVs isolated using exoEasy was not suitable for analysis using PEA [14]. In the current project, where “Buffer XE” was exchanged for PBS, PEA could be performed without any issues, suggesting that the buffer exchange was necessary. Recently, Görgens et al. demonstrated that EVs stored in PBS could lead to reduced recovery, and that PBS supplemented with human albumin and trehalose (PBS-HAT) is a superior storage buffer. For future studies, it might be beneficial to exchange the buffer to PBS-HAT to further improve the results [28], although the potential interference of the buffer with the proteomic approaches has to be evaluated first. One of the drawbacks of using saliva compared to blood samples is that the target analyte in some cases might have a lower concentration. The size of EVs enables not only buffer exchange but also concentration using ultrafiltration, and this might be one approach to overcome the potential low concentration of the target protein.

Using label-free quantitative proteomics, saliva-EVs and plasma-EVs were compared. Our results indicate that plasma-EVs had higher amounts of contaminating proteins and lower amounts of EV-related proteins, suggesting that saliva-EVs have a higher purity than plasma-EVs. The correlation between independent isolates (*n* = 3) was higher for saliva-EVs compared to plasma-EVs in this study. The correlation between the two types of EVs was only moderate, indicating that comparisons between the two groups have to be carried out carefully. We analyzed the 295 proteins that were upregulated in saliva-EVs using the STRING Database [29,30], and 290 proteins were successfully mapped. One of the top enriched Kyoto Encyclopedia of Genes and Genomes (KEGG) pathways were “salivary secretion” (14 proteins). This pathway was not upregulated in plasma-EVs (287 proteins mapped), giving us confidence in both EV-isolation and proteomic analysis. The KEGG pathway with the highest strength in plasma-EVs, was “Complement and coagulation cascades”, while for saliva-EVs it was “Proteasome” (Appendix A).

The developed workflow could be used for all procured individual samples (*n* = 10 ALS patients and *n* = 10 controls); however, a 0.8 µm filtration step was included as we sometimes noted clogging of the exoEasy membranes using other individual saliva samples (not included in the study). Some of the samples had visible contamination, such as gel-like appearance, brown particles or fibers. More volume of these samples was lost during the clearing steps compared to the other samples, but the downstream analyses were robust and there were no deviations in the results. Based on our experiments in this study with pooled saliva, we decided to set the minimum required input to two micrograms. Out of the 20 samples, the minimum EV yield was 2.6 µg EV protein per mL cleared saliva and the maximum loss was 62%. Consequently, a collected volume of only 1.25 mL saliva should be enough for this workflow, a volume that is easily collected from patients and donors.

In the current work, no significantly different proteins were detected between ALS patients and controls, in either saliva or in saliva-EVs. There was a trend towards the downregulation of ZNF428 in ALS patients. The function of this protein remains unknown, but it is a favorable prognostic marker in pancreatic cancer [31]. In saliva samples, there was a complete separation between patients and controls (upregulation in ALS patients) for the protein IGLL1, although the difference was not statistically significant after taking into consideration false discovery rate. IGLL1 is also called CD179B and is critical for B-cell development, but is expressed by many different cells. To the best of our knowledge, neither ZNF428 nor IGLL1 have been previously connected to ALS.

In conclusion, we have developed a workflow for the isolation of saliva-EVs followed by label-free quantitative proteomic analysis of the EVs and cleared saliva samples. Although the current study was underpowered, we could demonstrate the technical feasibility of the workflow by a successful analysis of twenty low-volume, individual samples. Future studies will include a higher number of samples for each group and validation using targeted analyses. We believe that the reported method can be useful for the detection of non-invasive biomarker candidates for many different diseases.

## 4. Materials and Methods

### 4.1. Pooled Saliva-EV Isolation

Pooled saliva was purchased from Lee Biosolutions Inc. (Maryland Heights, MO, USA) and used after approval from the Institutional Review Board placed in Takeda Pharmaceutical Company and Shonan Health Innovation Park. (No. CS-00200466). The saliva was cleared using differential centrifugation: 300× *g*, 10 min at room temperature, supernatant transferred and subjected to 3000× *g*, 10 min at room temperature. The supernatant was collected and used for EV-isolation. For bulk saliva-EV isolation, 90 mL cleared saliva was used for each run. EVs were isolated using exoEasy (Qiagen, Hilden, Germany). One milliliter buffer XE was used per column and applied twice to the membrane. The centrifugation steps were performed at room temperature. Ultrafiltration (Vivacon 2, 100 kDa, Sartorius, Göttingen, Germany) was used to exchange the buffer with PBS (Nacalai Tesque, Japan). PBS was added to reach a final volume of 1500 µL for each sample. We used 1.5 mL Protein LoBind tubes (Eppendorf, Hamburg, Germany) and stored the samples in a −80 °C freezer.

### 4.2. Individual Saliva Procurement and EV-Isolation

Saliva samples from ALS patients (*n* = 10) and controls (*n* = 10) were procured through National BioService (Saint Petersburg, Russia) in 2021, and used after approval from the Institutional Review Board placed in Takeda Pharmaceutical Company and Shonan Health Innovation Park (No. CS-00201029). Before collection, the participants were asked not to drink or eat anything but water within one hour, and no water for ten minutes before sample collection. The participants were then asked to rinse their mouths with water and wait another ten minutes before saliva collection (to avoid sample dilution). The participants were asked to allow saliva to pool under the tongue, and with the head tilted forwards gently guide saliva from the mouth into a centrifuge tube. The samples were aliquoted in 1 mL cryovials, frozen at −80 °C and shipped on dry ice to Takeda.

The samples were thawed and cleared by differential centrifugation (300× *g* for 5 min followed by 3000× *g* for 10 min, at room temperature). The supernatant was then subjected to 0.8 µm filtration using syringe filters (SLAAR33SS, Merck Millipore, MA, USA). EVs were then isolated using exoEasy (Qiagen, Hilden, Germany), 400 µL buffer XE was used and applied twice to each column. Buffer exchange was performed using ultrafiltration (Vivacon 500, 100 kDa filters, Sartorius, Göttingen, Germany). PBS was added to reach a volume of 10% of the starting volume. We used 1.5 mL Protein LoBind tubes (Eppendorf, Hamburg, Germany) and stored the samples in a −80 °C freezer.

### 4.3. Plasma EV-Isolation

Eight milliliters of pooled plasma (PLA151, Biopredic International, Saint-Grégoire, France) was used for plasma EV-isolation. The plasma was filtered through a 0.8 µm syringe filter (SLAAR33SS, Merck Millipore, MA, USA) and divided in three equal parts (each 1670 µL) for individual EV-isolation. EVs were isolated from the filtered plasma using exoEasy (Qiagen, Hilden, Germany) according to the manufacturer’s instructions. A total of 400 µL buffer XE was used and applied twice to each column. Buffer exchange was performed using ultrafiltration (Vivacon 500, 100 kDa filters, Sartorius, Göttingen, Germany). PBS was added to reach a volume of 10% of the starting volume. We used 1.5 mL Protein LoBind tubes (Eppendorf, Hamburg, Germany) and stored the samples in a −80 °C freezer.

### 4.4. Total Protein Quantification

Protein concentration was measured using NanoOrange (Invitrogen, Waltham, MA, USA), according to the manufacturer’s instructions. Pooled saliva-EVs were diluted 1:100, individual saliva-EVs 1:70, and cleared saliva 1:350.

### 4.5. Albumin Measurement

Albumin was measured using a commercially available ELISA kit (E88-129, Bethyl Laboratories, MA, USA). Pooled saliva-EVs were diluted 1:160 and cleared saliva 1:5120.

### 4.6. Nanoparticle Tracking Analysis

The EV size distribution and concentration were measured using nanoparticle tracking analysis (NanoSight NS500Z, Malvern Panalytical, Malvern, UK). The EVs were diluted 1:100 in PBS. Five thirty-second videos were recorded and analyzed for each sample, using Nanoparticle Tracking Analysis software version 2.3 build 0033 (Malvern Panalytical, Malvern, UK). We used the following capture settings: EMCCD camera type, shutter length of 20 ms, shutter setting 600, camera gain 250 and frame rate 21.2 fps. The following analysis settings were used: Background Extract: On, Detection Threshold: 10—Multi, Blur: Auto, Min track length: Auto, Min expected size: Auto, Temperature 29.0 °C, Viscosity 0.81 cP.

### 4.7. Transmission Electron Microscopy

Transmission electron microscopy was performed by Filgen (Nagoya, Japan). The EVs (2.5 µg) were diluted to 25 µL using PBS and subsequently mixed with 25 µL glutaraldehyde 0.1 molar phosphate buffer pH 7.4 (provided by Filgen). After fixation, the EVs were shipped on wet ice to Filgen. Then, 400 mesh copper grids and negative staining (2% uranyl acetate) was used and the EVs were observed under a Hitachi H-7600 microscope at 100 kV. EVs from three independent isolations were visualized, with pictures taken at 50,000× and 200,000× magnification.

### 4.8. Proximity Extension Assay

Ten microliters of sample were mixed with thirteen microliters M-PER (Thermo Fischer Scientific, Waltham, MA, USA), including a 1:100 protease inhibitor cocktail (Cell Signaling Technology, Danvers, MA, USA). The samples were shipped on dry ice for proximity extension assay by Olink Proteomics (Waltham, MA, USA) using the panel “Olink Target 96 Cardiovascular III (v.6113)”.

### 4.9. Label-Free Quantitative Proteomics

The samples were shipped on dry ice to PhenoSwitch Bioscience (Sherbrooke, Canada). The samples were reduced for 10 min at 50 °C with 10 mM DTT. The following amounts of protein were analyzed: for comparing pooled plasma-EVs and pooled saliva-EVs: 80 µg; for the dilution experiment: 20, 2 and 0.5 µg pooled saliva-EVs; for individual saliva: 27 µg; and for individual saliva-EVs: 4 µg. The samples were acidified with 2% formic acid and the peptides were purified by reversed phase SPE. To generate an ion library, proteins from each sample were pooled and separated on a 4–12% SDS PAGE and cut into 12 individual fractions. Each fraction was reduced, alkylated and digested in the gel. After digestion, peptides were extracted from the gels by sonication and purified by reversed phase SPE.

Acquisition was performed with a Sciex TripleTOF 6600 (Sciex, Foster City, CA, USA) equipped with an electrospray interface with a 25 µm iD capillary and coupled to an Eksigent µUHPLC (Eksigent, Redwood City, CA, USA). Analyst TF 1.8 software was used to control the instrument and for data processing and acquisition. Acquisition was performed in Information Dependent Acquisition (IDA) mode for the 12 fractions from the pool. Separation was performed on a reversed phase Kinetex XB column 0.3 µm i.d., 2.6 µm particles, 150 mm long (Phenomenex), which was maintained at 60 °C. Samples were injected by loop overfilling into a 5 µL loop. For the 45 min (IDA) and 45 min (SWATH) LC gradient, the mobile phase consisted of the following solvent A (0.2% *v*/*v* formic acid and 3% DMSO *v*/*v* in water) and solvent B (0.2% *v*/*v* formic acid and 3% DMSO in EtOH) at a flow rate of 3 µL/min.

Proteins for the ion library were identified using the 2D LC-MS/MS files in Protein Pilot. Proteins were quantified against a combined ion library (Swath Atlas and home-made ion library) in the SWATH 2.0 applet from the Peakview software. Quantification was carried out using 10 peptides per protein, 4 MS/MS ions per peptide, 12.5 min retention time window and 25 ppm XIC width. Peptides for every sample were then corrected using an in-house script with the RT-LOESS algorithm from the NormalizerDE [32] package. After correction, peptides for every protein were summed and reported.

### 4.10. Data Analysis

Data analysis was performed using GraphPad Prism 9.4.1 (GraphPad Software, San Diego, CA, USA). All error bars represent standard deviation. Multi scatter plots, volcano plots, principal component analyses and heat maps were produced with Perseus (version 1.6.15.0) [33] using default settings.

## Figures and Tables

**Figure 1 ijms-24-05237-f001:**
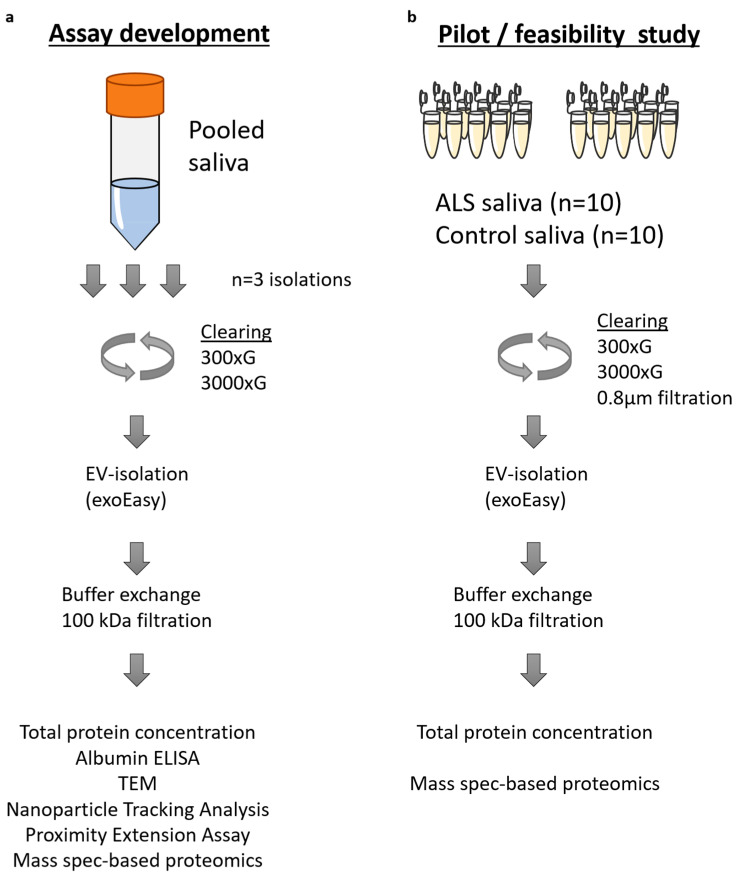
Study overview. (**a**) Pooled, healthy control saliva was used for assay development (90 mL × 3 isolates). The saliva was cleared using differential centrifugation followed by isolation of extracellular vesicles using exoEasy. The isolates then underwent buffer exchange using ultrafiltration. The EVs were analyzed for total protein concentration, albumin concentration, transmission electron microscopy, proximity extension assay and mass spectrometry-based proteomics. (**b**) As a feasibility and pilot study, amyotrophic lateral sclerosis and matched control saliva samples were used (*n* = 10 each) with a volume ranging from 2.1 to 4.9 mL. The isolation protocol was the same except for minor changes in the clearing step. Isolated EVs and whole saliva samples were analyzed using total protein concentration measurements and mass spectrometry-based proteomics.

**Figure 2 ijms-24-05237-f002:**
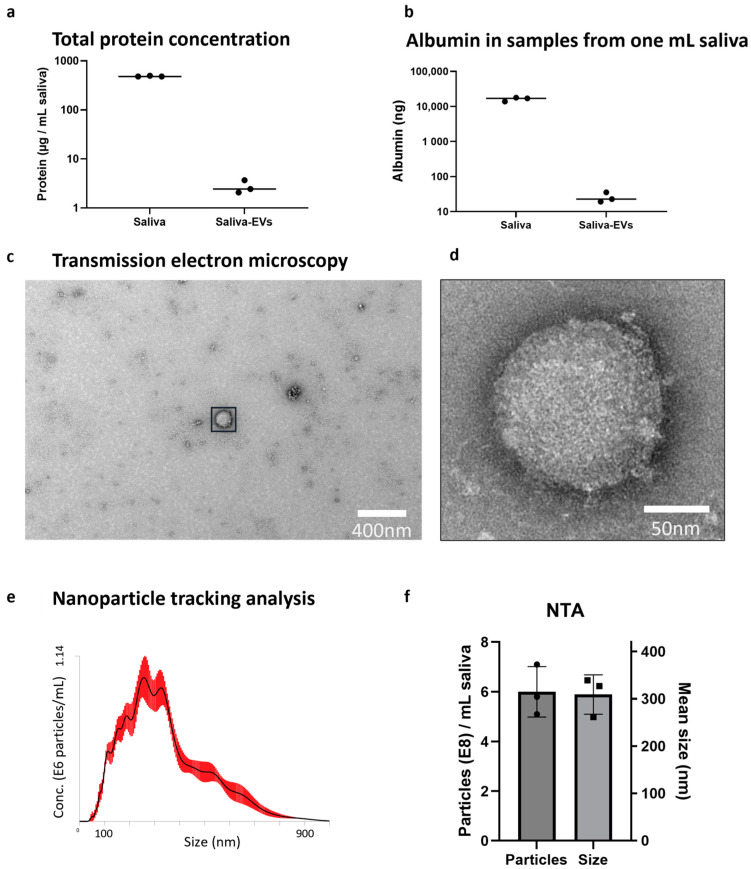
Characterization of saliva-EVs. Both total protein concentration (**a**) and albumin (**b**) were considerably reduced in the saliva-EVs compared to whole saliva. Transmission electron microscopy identified spherical structures at low (**c**) and high magnification (**d**). (**e**) Nanoparticle tracking analysis was used to measure vesicle concentration and size distribution. (**f**) One milliliter of saliva yielded around 6 × 10^8^ particles with a mean size of approximately 300 nm. Error bars represent standard deviation.

**Figure 3 ijms-24-05237-f003:**
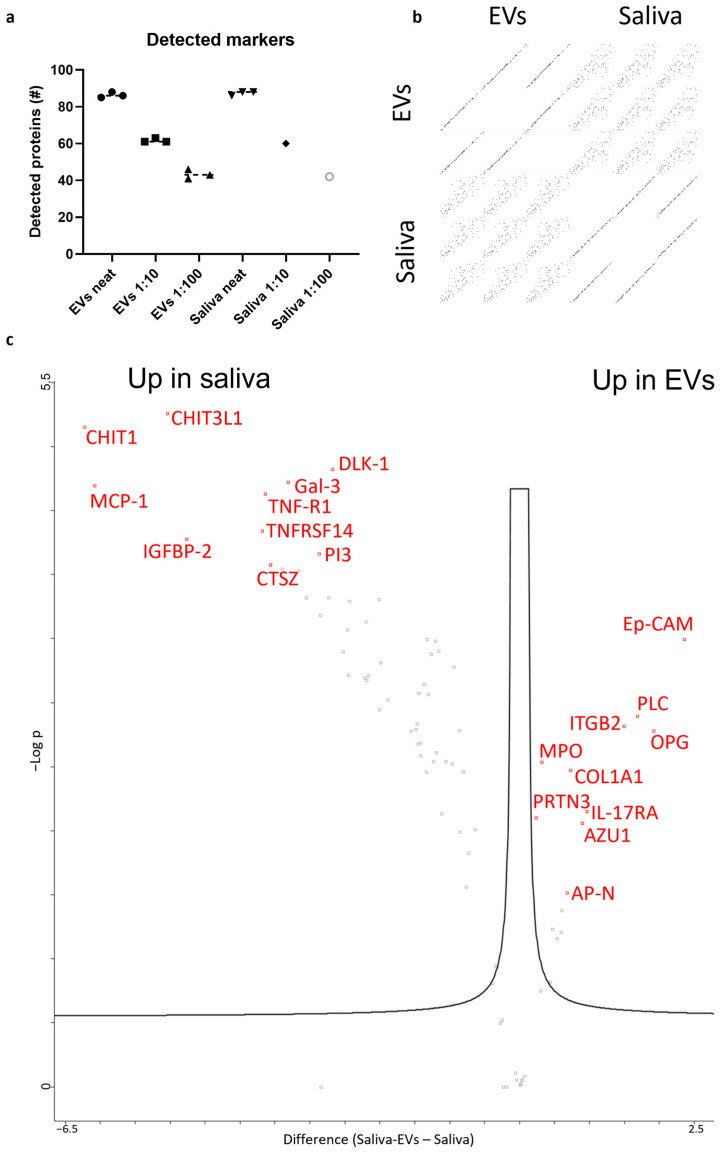
Proximity extension assay of saliva and saliva-EVs. (**a**) Scatter plot indicating the number of detected proteins in different dilutions of saliva and saliva-EVs. (**b**) Both saliva and saliva-EV samples showed a high correlation between samples within each group, while the inter-group correlation was lower. (**c**) A volcano plot demonstrating differences between the groups. The top 10 differentially expressed proteins in each group are labelled.

**Figure 4 ijms-24-05237-f004:**
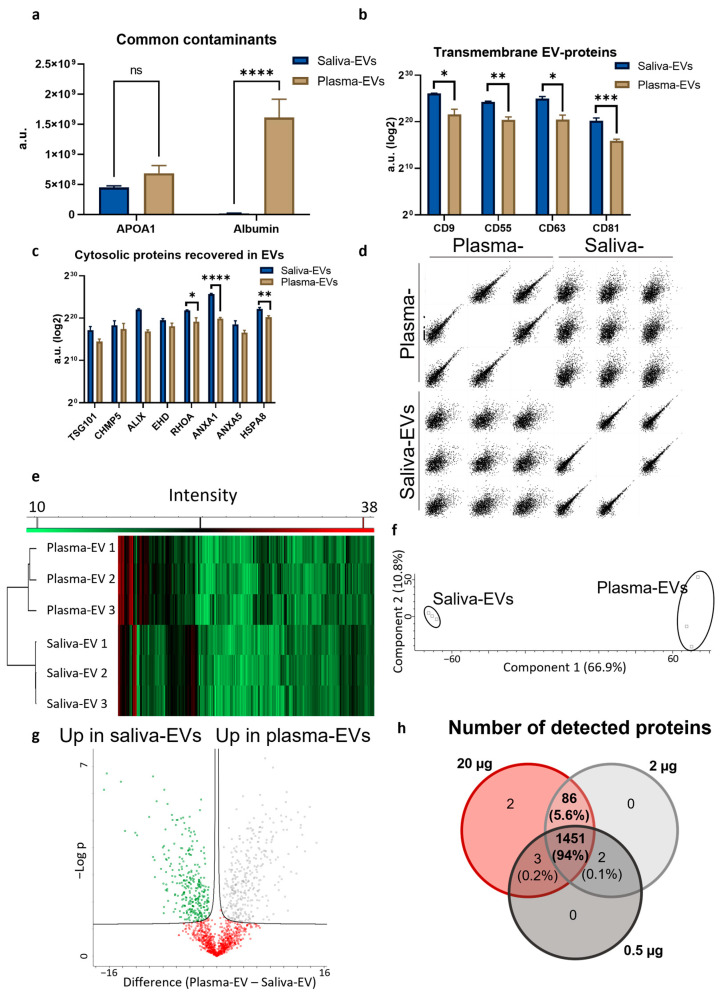
Mass spectrometry-based proteomics of saliva- and plasma-EVs. (**a**) We compared saliva-EVs to plasma-EVs and found that albumin, a contaminating protein, was more abundant in plasma-EVs, while there was no significant difference in APOA1. Several common transmembrane (**b**) and cytosolic (**c**) EV-associated proteins showed a higher expression in saliva-EVs compared to plasma-EVs. (**d**) A multi-scatter plot of saliva and plasma-EVs indicates strong intra-group correlations which are higher in the saliva-EVs compared to plasma-EVs. Unsupervised hierarchical clustering (**e**) and principal component analysis (**f**) could correctly cluster all samples. (**g**) A volcano plot was generated, depicting proteins which were upregulated in saliva-EVs (green squares), upregulated in plasma-EVs (gray squares) or with similar expression between the two groups (red squares). (**h**) To determine the minimum required amount of total protein for the detection of sufficient proteins, 0.5 µg, 2 µg and 20 µg of protein were analyzed. Two micrograms protein were deemed sufficient, with a minimal difference to 20 µg. * *p* = 0.05 ** *p* = 0.01, *** *p* = 0.001, **** *p* = 0.0001, ns—non-significant by two-sample *t*-test with permutation-based FDR. Error bars represent standard deviation.

**Figure 5 ijms-24-05237-f005:**
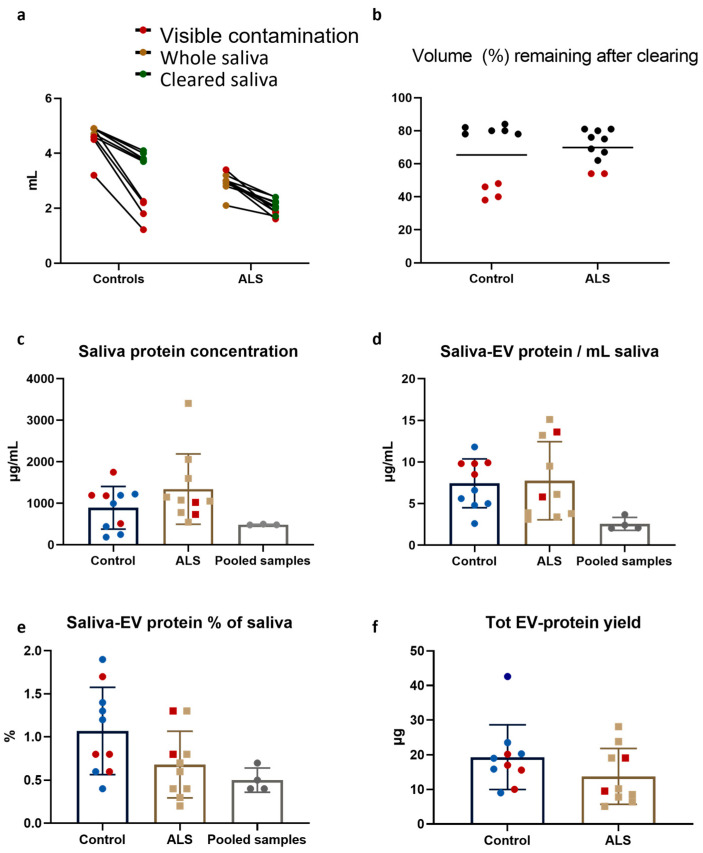
EV-isolation from patient and control saliva samples. (**a**) The starting volume and volume of cleared saliva for controls and ALS samples are plotted. Yellow dots represent whole saliva, green dots cleared saliva and red dots saliva with visible contamination. (**b**) Approximately 65% and 70% of the starting volume remained after clearing of control and ALS samples, respectively. A greater reduction was observed for contaminated samples (red dots) compared to non-contaminated samples (black dots). (**c**,**d**) The total protein concentrations in saliva (**c**) and saliva-EVs isolated from one milliliter saliva (**d**) were similar between the two groups, and within the range of pooled samples. (**e**) The EV-isolates represented approximately 0.75% of the total saliva protein concentration, with no significant difference between the sample groups and within the range of pooled samples. (**f**) The total EV-protein yield ranged from 42.6 µg to 5.1 µg, with no difference between the groups. For (**c**–**f)**: blue dots represent control samples, yellow squares ALS samples and gray dots pooled samples. Red dots or squares indicate samples with visible contamination. Error bars represent standard deviation.

**Figure 6 ijms-24-05237-f006:**
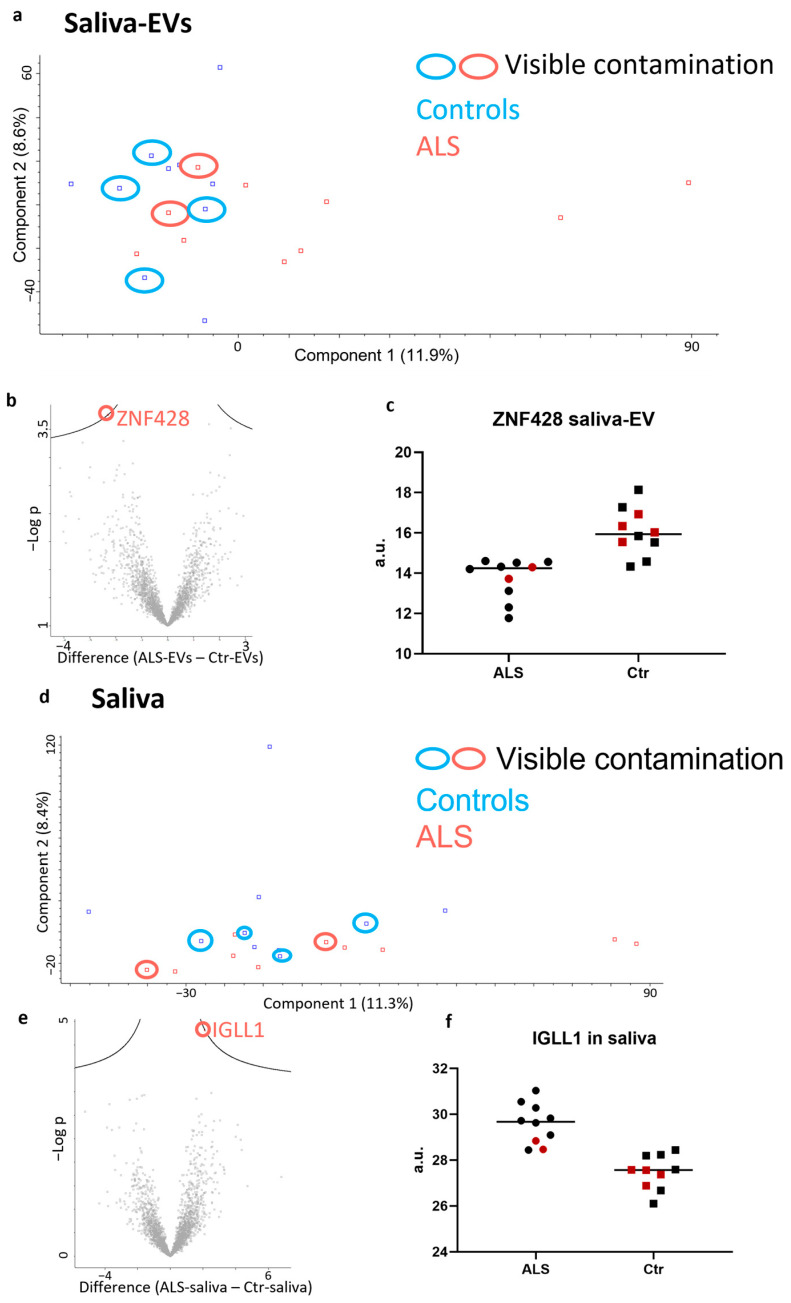
Proteomic analysis of saliva-EVs and cleared saliva. (**a**) There was no clear clustering of the samples based on principal component analysis and the contamination did not appear to affect the results. (**b**) There was no statistically significantly expressed protein between the groups, as depicted in a volcano plot, but there was a trend towards downregulation of ZNF428 in ALS saliva-EVs (**c**) black dots represent ALS samples, black squares control samples and red dots or squares represent samples with visible contamination. (**d**) Similarly, for saliva samples, there was no obvious clustering of the samples based on principal component analysis. (**e**) No proteins were statistically significantly expressed, although IGLL1 showed a complete separation between patients and controls (**f**) black dots represent ALS samples, black squares control samples and red dots or squares represent samples with visible contamination.

**Table 1 ijms-24-05237-t001:** Saliva donor details and clinical characteristics.

Age	Gender	Diagnosis	Co-Morbidity	Treatment
64	M	Sporadic ALS with bulbar and pseudobulbar abnormalities in the form of dysarthria, dysphagia, quadraparesis	-	Thiamine 50 mg per day I.M., Pyridoxine 100 mg per day I.V., Cyanocobalamin 1 mg per day I.V.
62	M	Sporadic ALS in the form of ambilateral pyramidal signs, bulbar abnormalities	IHD, Stenocardia	Thiamine 100 mg per day I.M., Pyridoxine 150 mg per day I.V., Cyanocobalamin 1 mg per day I.V, Simvastatin 20 mg/day, Enalapril 20 mg/day, Aspirin cardio 150 mg per day, Nitrospray 0,4 mg on demand.
60	F	Sporadic ALS in the form of spastic tetraparesis with walking dysfunction and pseudobulbar syndrome	Lumbodynia	Thiamine 50 mg per day I.M., Pyridoxine 100 mg per day I.V., Cyanocobalamin 1 mg per day I.V.
70	M	Sporadic ALS in the form of multimodal paraparesis	IHD, Effort angina.	Thiamine 50 mg per day I.M., Pyridoxine 100 mg per day I.V., Cyanocobalamin 1 mg per day I.V, Simvastatin 10 mg/day, Enalapril 10 mg/day, Aspirin cardio 100 mg per day, Nitrospray 0.4 mg on demand.
54	F	Sporadic ALS in the form of moderate distal paraparesis, fasciculations, moderate distal amyotrophias	IHD	Thiamine 50 mg per day I.M., Pyridoxine 100 mg per day I.V., Cyanocobalamin 1 mg per day I.V, Enalapril 20 mg per day
57	M	Sporadic ALS in the form of spastic tetraparesis with walking dysfunction and pseudobulbar syndrome	Lumbodynia	Thiamine 100 mg per day I.M., Pyridoxine 150 mg per day I.V., Cyanocobalamin 1 mg per day I.V.
71	M	Sporadic ALS in the form of moderate distal paraparesis, fasciculations, moderate distal amyotrophias	IHD	Thiamine 50 mg per day I.M., Pyridoxine 100 mg per day I.V., Cyanocobalamin 1 mg per day I.V, Enalapril 20 mg per day
65	F	Sporadic ALS in the form of multimodal paraparesis	Stenocardia	Thiamine 50 mg per day I.M., Pyridoxine 100 mg per day I.V., Cyanocobalamin 1 mg per day I.V, Rosuvastatin 30 mg per day, Enalapril 20 mg per day
32	F	Sporadic ALS in the form of multimodal paraparesis		Thiamine 50 mg per day I.M., Pyridoxine 100 mg per day I.V., Cyanocobalamin 1 mg per day I.V.
46	M	Sporadic ALS in the form of ambilateral pyramidal signs, bulbar abnormalities	IHD, Stenocardia	Thiamine 50 mg per day I.M., Pyridoxine 100 mg per day I.V., Cyanocobalamin 1 mg per day I.V, Aspirin cardio 100 mg per day, Enalapril 20 mg/day
64	M	Control	-	-
65	M	Control	IHD	Thiamine, Pyridoxine, Cyanocobalamin, Rosuvastatin 30 mg/day, Enalapril 20 mg/day.
68	F	Control	Chronic gastritis	Thiamine, Pyridoxine, Cyanocobalamin, Omeprazole 20 mg per day.
56	F	Control	IHD, Stenocardia	Thiamine, Pyridoxine, Cyanocobalamin, Simvastatin 10 mg/day, Enalapril 10 mg/day, Aspirin cardio 100 mg per day, Nitrospray 0.4 mg on demand.
74	F	Control	Chronic gastritis	Thiamine, Pyridoxine, Cyanocobalamin, Pantoprazole 20 mg 2 times per day.
70	F	Control	Chronic gastritis	Thiamine, Pyridoxine, Cyanocobalamin, Pantoprazole 20 mg 2 times per day.
63	F	Control	-	-
32	F	Control	-	-
44	M	Control	-	-
63	F	Control	-	-

## Data Availability

Data, including proteomics raw data, will be available upon reasonable request.

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
