# Peer review of "Saliva and Saliva Extracellular Vesicles for Biomarker Candidate Identification—Assay Development and Pilot Study in Amyotrophic Lateral Sclerosis"

_ijms, 2023, doi:10.3390/ijms24065237_

Round 1
Reviewer 1 Report (Previous Reviewer 2)
Authors have made revision to the manuscript as per the reviewer comments. Now the manuscript is deemed to be acceptable for publication.
Author Response
We thank the Reviewer for his or her work.
Reviewer 2 Report (Previous Reviewer 3)
The authors addressed my comments and improved the manuscript. Description of the patients was included in the text and legibility of figures improved.
Author Response
We thank the Reviewer for his or her work.
Reviewer 3 Report (New Reviewer)
The authors describe their strategy to isolate and characterize EVs from saliva with different proteomic platforms.
The work is a nice effort but fail to provide statistical significant results when comparing EVs from ALS patients to Control samples.
The authors have a PEA dataset that could be compared to the Mass spectrometry generated dataset. Is there any overlap in identified/quantified proteins?
I don't believe that it is relevant to compare serum versus saliva EV datasets since the serum dataset contain all the serum proteins in high abundance. These contaminants could be discussed. Is it a problem of the kit used for the EV isolation?
Can the existence of the co-morbidities influence the data analysis/interpretation? Please discuss.
Are the two suggested protein candidates ZNF428 and IGLL1 identified with more than one peptide? Can the authors provide the peptide information as supplementary material?
The volumes/amounts of starting material is a bit confusing. How many ml initial saliva are used in the individual patient / control MS analyses?
Finally, is the dataset uploaded to any open database like PRIDE?
Author Response
Please see the attachment

This manuscript is a resubmission of an earlier submission. The following is a list of the peer review reports and author responses from that submission.
Round 1
Reviewer 1 Report
Dear Authors,
The manuscript entitled “Saliva and saliva extracellular vesicles for biomarker candidate identification-assay development and pilot study in amyotrophic lateral sclerosis” developed methods for saliva biomarker identification using EV isolation and proteomic evaluation. This study is interesting, however, the methods used in this study can not obtain high purity EV from saliva. For the proteomic analysis of EV, purity is a crucial factor. Another problem is that the manuscript is poorly written. Many experimental details are missing, and grammar errors are throughout the manuscript. I suggest to reject the manuscript. Please see my comments and suggestions below.
Comments and suggestions
1. Line 11, correct “that” to “those”.
2. Line 54-61, the authors introduced what is EVs. The background about saliva derived EVs needs to be briefly introduced. For example, which cells contribute to EVs in saliva.
3. The authors mentioned that saliva is easily contaminated with viruses, bacteria, and food debris. How this issue can be improved needs to be briefly introduced.
4. Line 55, correct “an essential role” to “essential roles”.
5. Figure 1, saliva is viscose. Do the authors think cells or cell debris can be removed efficiently by spinning at 300 g followed by 3000 g without diluting with PBS?
6. Figure 1a, the authors mentioned in the introduction, saliva contains viruses, bacteria and food debris. Can the protocol reported in this study isolate high purity EV from saliva?
7. Figure 2e, the NTA data shows that the isolated saliva-EV is big. These data is not consistent with the typical size of EVs. Also, figure 2e is controversial to the TEM data in figure 2d. In figure 2d, the size of the majority particles looks very small.
8. Did the authors do western blotting to test EV markers? For instance, CD9, CD63 or CD81.
9. Line 75, correct “was” to “were”.
10. Line 121-122, this sentence is hard to understand.
11. Figure 4, how the plasma EVs were isolated?
12. Why albumin was more abundant in saliva EVs? Plasma should contain more albumin.
13. Figure 6, the data needs to be validated using western blotting.
14. Line 271, correct “in room temperature” to “at room temperature”.
15. Line 279, in a -80 °C freezer.
16. Please check the space between number and unit. For instance, Line 291.
17. Line 307, correct EVs to EV.
18. Line 206, the parameters of NTA need to be clarified, for instance, camera level, length of each video, etc.
19. Line 314, what reagent was used for negative staining before TEM imaging?
20. Line 328, how the proteomic data was analyzed? How do the authors set thresholds to filter the data?
21. Please check the manuscript thoroughly, grammar errors are throughout the manuscript.
Reviewer 2 Report
Authors have carried out nice piece of work on " Saliva and saliva extracellular vesicles for biomarker candidate identification – assay development and pilot study in amyo- trophic lateral sclerosis”. The manuscript is extremely well written, easy to comprehend. Authors have explored methodologies for analyzing saliva and saliva extracellular vesicles. The information provided in this manuscript useful to the researchers, adds a value to the scientific community.
I have few points to improve the manuscript
1. In the introduction section, authors would require to mention previous works and literature on characterization of saliva and saliva extracellular vessels if any.
2. In page 4, line 92, the mean particle size for isolated extracellular vesicles was 309±41 nm, which is on the higher size. For most of the EVs, the particle size is reported below 100 nm. Please justify for obtaining higher size EVs in your study.
3. In the conclusion section, authors mentioned that “Future studies will include a higher number of samples for each group. Will there any additional characterization studies with increase in sample size?
Reviewer 3 Report
The manuscript describes workflow for biomarker identification in saliva extracellular vesicles. Utilization of saliva EVs for diagnostic purposes is an attractive topic due to noninvasivness of saliva collection and its possible utilization in diagnosis of number of different pathologies including neurodegenerative diseases. The concept of the study is quite straightforward and utilizes commercially available biological materials from different sources (saliva samples, plasma samples), affinity purification of EVs on ExoEasy Qiagen spin columns, characterization of EVs fractions and their proteomic analysis. Majority of the analytical procedures was carried out in commercial laboratories. The main finding of the study is that the procedure led to identification of approximately 1500 proteins in isolated EVs fractions and is thus suitable for its use in a new biomarker identification. Weak point of the study is the absence of significant differences found in a pilot study of ALS patients and control group. The description of the subjects included in this pilot study is also missing, the Table 1 which is supposed to contain patients characteristics is not present. While the authors provide TEM images of EVs, their count by NTA and show the presence of several EVs markers in the isolated fractions it might be also useful to check the obtained proteomic data against the ExoCarta database to document further the EVs presence. The text is well written, but figures need improvements, many fonts used are too small to be legible.
Specific comments
1) Fig. 3 - too small fonts used in b) and c)
2) Line 132 - The text should be: Saliva-EVs were compared to plasma-EVs....
3) Fig. 4 - too small fonts used in d), e), f) a g)
4) Fig. 6 - too small fonts used in a), b), d) and e)
5) Line 176 - Table 1 is missing
Round 2
Reviewer 1 Report
Dear Authors,
Thank you for carefully revising the manuscript entitled “Saliva and saliva extracellular vesicles for biomarker candidate identification-assay development and pilot study in amyotrophic lateral sclerosis”. Some of my concerns are responded, however, crucial data are still missing. Also, some results are controversial. Some sentences are hard to understand, thus language editing is needed. The manuscript does not meet the high standard of IJMS. I suggest to reject the manuscript.
Comments and suggestions:
1. The EV isolation method described in this study is not good enough. Saliva is a kind of high viscosity fluid. To remove cells or collect EVs efficiently, dilution of saliva is strongly recommended. See following published papers:
https://bmccancer.biomedcentral.com/articles/10.1186/s12885-018-4364-z
https://www.nature.com/articles/s41598-021-87180-4
2. Fig 2c and 2d are controversial to fig 2e. NTA data shows that the isolated EV have a broad size range. However, in fig 2c, only two particles are shown. Also, the shown particles do not have classical EV structure and morphology. I suggested the authors to repeat figure 2c and 2e.
3. The authors stated that MS data shows that CD9, CD63 and CD81 are present. According to MISEV 2018, western blotting to detect EV markers is needed.
4. Figure 6, validation of ZNF428 and IGLL1, for instance using western blotting, is needed.
5. The manuscript is not well written, English editing is needed.